# Pittsburgh Sleep Quality Index (PSQI) responses are modulated by total sleep time and wake after sleep onset in healthy older adults

**Jennifer Zitser**[1,2,3], **Isabel Elaine Allen**[1,2,4], **Neus Falgàs**[1,2], **Michael M. Le**[1], **Thomas C. Neylan**[1,5,6], **Joel H. Kramer**[1,2,5], **Christine M. Walsh**[1,2] *

1 Memory and Aging Center, Department of Neurology, Weill Institute for Neurosciences, University of California, San Francisco, CA, United States of America, 2 Global Brain Health Institute, San Francisco, CA, United States of America, 3 Movement Disorders Unit, Department of Neurology, Tel Aviv Sourasky Medical Center, affiliated to the Sackler Faculty of Medicine, Tel-Aviv University, Tel-Aviv-Yafo, Israel, 4 Department of Biostatistics and Epidemiology, University of California, San Francisco, CA, United States of America, 5 Stress & Health Research Program, Department of Mental Health, San Francisco VA Medical Center, San Francisco, CA, United States of America, 6 Department of Psychiatry, Weill Institute for Neurosciences, University of California, San Francisco, CA, United States of America

* Christine.Walsh@ucsf.edu

## Abstract

### Objectives

To investigate the objective sleep influencers behind older adult responses to subjective sleep measures, in this case, the Pittsburgh Sleep Quality Index (PSQI). Based on previous literature, we hypothesized that SE would be associated with PSQI reported sleep disruption. Furthermore, because SOL increases progressively with age and it tends to be easily remembered by the patients, we also expected it to be one of the main predictors of the perceived sleep quality in the elderly.

### Methods

We studied 32 cognitively healthy community-dwelling older adults (age 74 ± 0.3 years) who completed an at-home sleep assessment (Zeo, Inc.) and the PSQI. Linear mixed models were used to analyze the association of the objective sleep parameters (measured by the Zeo) with the PSQI total score and sub-scores, adjusting for age, gender, years of education and likelihood of sleep apnea.

### Results

Objective sleep parameters did not show any association with the PSQI total score. We found that objective measures of Wake after sleep onset (WASO, % and min) were positively associated with the PSQI sleep disturbance component, while SE and Total Sleep Time (TST) were negatively associated with PSQI sleep disturbance. Lastly, objective SE was positively associated with PSQI SE.

**Data Availability Statement:** Raw data that supports the findings of this study were generated at the Memory and Aging Center (MAC) at UCSF.

UCSF has always supported data sharing once we believe it is fundamental for the scientific community growth and the improvement of patient care. As well, we are committed to share the data consistent with applicable laws and regulations and the IRB's approval. Therefore, data from the current study will be available to external researchers after submitting the standard UCSF request online (see link below). https://ucsf.co1.qualtrics.com/jfe/form/SV_1ZazyeJGOE2Dbvv?Q_JFE=qdgh.

**Funding:** : This study was supported by UCSF Hillblom Aging Network grant, Clinical features and neuropathological basis of sleep-wake behavior in Alzheimer's Disease and PSP (NIA R01 AG060477), Linking sleep dysfunction to tau-related degeneration across AD progression (NIA R01 AG064314), Biological predictors of brain aging trajectories (NIA R01 5R01AG032289-10) and the Tau Consortium/Rainwater Charity Foundation.

**Competing interests:** The authors have declared that no competing interests exist.

## Conclusions

Our findings showed that WASO, SE and TST, are associated with PSQI sleep disturbance, where the greater WASO, overall lower SE and less TST, were associated with increased subjective report of sleep disturbance. As expected, subjective (PSQI) and objective measures of SE were related. However, PSQI total score did not relate to any of the objective measures. These results suggest that by focusing on the PSQI total score we may miss the insight this easily administered self-report tool can provide. If interpreted in the right way, the PSQI can provide further insight into cognitively healthy older adults that have the likelihood of objective sleep disturbance.

## 1. Introduction

Sleep disturbances are a common health complaint, with a prevalence that is expected to increase and have more important consequences in the elderly [1]. The National Sleep Foundation's 2003 Sleep in America Poll confirmed that 46% of adults aged 65–74 reported insomnia symptoms, while 39% napped. These rates increased to 50% and 46%, respectively, in participants aged 75–84 years [2]. Others have shown higher rates, with approximately 65% of adults over 65 years old reporting insomnia symptoms [3]. These sleep disturbances have been explained by age-related changes in sleep need and sleep architecture [4], including more sleep fragmentation, earlier awakening, less slow-wave sleep, and advanced circadian rhythms [5–7].

Assessment of insomnia and other sleep disturbances should include a structured clinical interview, sleep diary and objective measures like actigraphy or polysomnography (PSG) [8]. Yet in practice, resources -especially time and expertise- are scarce; thus self-report questionnaires are more frequently used than other forms of appraisal [9]. Diagnosing patients through low-cost tools (such as validated questionnaires) gives us the opportunity to treat patients with sleep disorders, reduce the likelihood of nursing home placement and improve the patient's and patient's family quality of life. However, it is important to understand the exact information that these subjective measures are giving us. The concordance of such scales with objective measures therefore warrants further investigation.

Among the rating scales commonly used, the Pittsburgh Sleep Quality Index (PSQI) was developed to measure general sleep quality -a construct that is still not well defined-[10] and is one of the recommended questionnaires for the study of global sleep and insomnia symptoms [8, 11].

If the PSQI validly reflects sleep, it should align closely with objective measures. However, analysis by age was not performed in the initial PSQI validation study, which is why since then, validation studies expanded to the aging and other different populations using PSG or actigraphy as objective measures, have been performed to address these potential correlates [12]. The importance of such validations is highlighted by the fact that, often times, the perception of sleep quality by older subjects differs from their actual sleep structure [13–16] and as a result, sleep neuroscientists, question the validity of subjective tools. Furthermore, while aging results in significant changes in sleep, it does not necessarily result in increased complaints about sleep quality [17, 18]. In fact, Buysse et al., suggested that despite their objectively disturbed sleep, healthy older individuals seldom complained, and they argued that the elderly tend to adapt their perception of what is "acceptable" sleep [14]. In other words, even though

the PSQI has been well validated to differentiate between poor and good sleepers in different populations, when PSQI was given to healthy younger and older subjects, despite the fact that objective sleep quality was worse among the elders, their global PSQI scores were in the 'good' sleep quality range [14]. In the same study, PSG-based objective sleep measures did not correlate with either global or component scores on the PSQI in the older adult cohort [14]. Similarly, Landry et al. found that older participants tended to have greater sleep fragmentation (as measured by actigraphy), yet reported shorter latency and fewer awakenings in their sleep diaries [19]. However, they also found that more individuals were "under-estimators" of their sleep quality, compared to "over-estimators", which suggests the opposite, that older adults might tend to perceive their sleep as worse than it actually might be. Others, also did not report a relationship between subjective ratings of sleep quality and PSG-derived measures [20] and similar results were found comparing PSQI and other subjective methods to actigraphy [21–23]. On the other hand, a positive correlation between PSG-derived sleep onset latency (SOL) and sleep efficiency (SE), was found with the PSQI total score of older men, but not women [16].

In summary, perceived sleep quality appears to be something quite different from objective sleep quality, and the reasons for these disparities -between objective and subjective measures- are unclear. It might be that limiting the acquisition of objective data to 1–2 nights (as opposed to this current study where we measure up to 10 nights) has limited previous work. Furthermore, in some previous studies data was acquired at an overnight sleep lab, which might not be a truthful representation of a person's sleep, in contrasts we collected data at home. It is also possible that the cohort assessed across studies were screened using different tools to determine if they were cognitively healthy older adults.

The main goal of the present study was to investigate the specific objective drivers behind older adult responses to subjective measures in the PSQI. Because of the frequent use of sleep quality ratings in sleep research and treatment (in fact, for many studies, the PSQI is the only measure used to quantify sleep quality), it will be helpful to understand what objective sleep factors predict a cognitively healthy older adult's perception of how well they sleep. Based on limited prior research available, we hypothesized SE would be associated with PSQI reported sleep disruption. Furthermore, because SOL increases progressively with age and it tends to be easily remembered by the patients, we also expected it to be one of the main predictors of the perceived sleep quality in the elderly.

## 2. Methods

### 2.1. Participants

Thirty-two community-dwelling older adults (age 74±0.3) were recruited from the longitudinal Hillblom Healthy Aging Network at the University of California, San Francisco Memory and Aging Center. This is an ongoing study where participants complete comprehensive evaluations at approximately 15-month intervals. Following a neurological exam, neuropsychological assessment, and an informant interview, all participants were determined to be cognitively healthy by a formal committee comprised of a board-certified neurologist and a neuropsychologist. Only participants who completed an at-home sleep study between August 2011 and January 2018, and had a PSQI and ZEO evaluation within a 1-month period were included in the study. Furthermore, participants had to be functionally intact as operationalized by a 0 on the Clinical Dementia Rating (CDR) scale via interviews with study partners, had no major memory concerns or diagnosed memory condition, and did not meet consensus criteria for any neurodegenerative disease. Each participant provided written, informed consent. The consent form and study protocol were approved by the UCSF Committee on Human Research.

## 2.2. Questionnaires

Participants completed the Pittsburgh Sleep Quality Index (PSQI). The PSQI was developed by Buysse et al. [11] and is a self-report assessment tool that evaluates sleep quality over a one-month period. A global score and seven component scores can be derived from the scale. The component scores are the following: Subjective sleep quality, sleep latency, sleep duration, sleep efficiency, sleep disturbances, use of sleeping medications and daytime dysfunction. Each component is scored on a scale from 0–3, with the total score ranging from 0–21; where a higher score describes poorer sleep quality. A total PSQI score greater than 5 has been validated as being highly sensitive and specific in distinguishing good from poor sleepers across a number of populations [24], including the elderly [11].

## 2.3. At-Home sleep assessment

The at-home sleep protocol was previously described in detail [25] and the at-home sleep assessment device is described more fully in previous papers [26, 27]. Participants wore a wireless sleep-monitoring device (Zeo, Inc.) for up to 10 nights (6.8 ± 2.2, range 3–10 nights) during their attempted sleep period. The headband has three dry electrodes at approximately Fp1, Fpz and Fp2. Electrophysiological signals are processed in 30 second epochs using a proprietary neural network (Zeo, Inc.) and assesses: latency to sleep onset (SOL), duration of wake after sleep onset (WASO), light sleep (stages 1 and 2 Non-REM sleep), deep sleep (stage 3 Non-REM sleep) and REM sleep.

The first night of data acquisition was used as a habituation night and removed from the dataset. Any subsequent night with 45 mins or more of unscored data was removed from the dataset. Light sleep, deep sleep and REM sleep were analyzed as both duration (mins) and as the percent of total sleep. Latency to sleep onset (SOL) was calculated as the number of minutes until the first epoch of sleep. WASO was the number of minutes spent awake between the first and last epoch of sleep. Sleep efficiency (SE) was calculated as the percentage of time spent asleep divided by the total time in bed trying to sleep. Previous papers have shown relatively high levels of agreement between light sleep, deep sleep and REM sleep measured on PSG and via Zeo [26, 27]. Given the long recruitment period it is important to highlight that there were no commercial updates in the ZEO software during the time of the study and the ZEO measures were held constant throughout. Sensors and headbands were replaced by newer ones if needed.

# 3. Statistical analysis and results

## 3.1 Statistical analysis

Demographic characteristics and outcomes of subjective sleep data are presented as means (standard deviation) or frequencies (percentages). Linear mixed models were used to analyze the effect of objective sleep parameters on PSQI subjective sleep outcomes adjusting for age, gender, years of education, Berlin Sleep Questionnaire (an assessment of the likelihood of sleep apnea) [28] and variability of each objective sleep parameter (standard deviation). A random effect for events within subject was used as subjects contributed a varying number of sleep events. Models were fit separately for objective sleep measures and their standard deviations due to the small sample size. All statistical analyses were conducted using Stata/SE 16.1 (College Station, Texas, USA).

## 3.2 Results

**a) Demographics and sleep data.** Demographic and clinical data are shown in Table 1. The sample had a mean age of 74 (67–84 years old) and was predominantly female (59.4%).

**Table 1. Demographics and subjective sleep scores.**

|  | Healthy subjects (n = 32) |
|---|---|
| **Demographics** |  |
| Age | 74±0.3 (Range 67–84) |
| Gender (% women) | 59.4% |
| Years of education | 18.1±1.7 |
| MMSE | 29.4±1 |
| **Subjective sleep scores** |  |
| PSQI total score | 4.4±3.2 |
| Berlin Apnea (% low risk) | 81.25% |
| **Objective sleep scores (Zeo)** |  |
| WASO (Min) | 46.4±33.5 |
| Light Sleep (%) | 61.7±8.8 |
| Deep Sleep (%) | 9.3±5 |
| REM Sleep (%) | 28.4±9.7 |
| Sleep Onset Latency (Min) | 35.6±22.3 |
| SE (%) | 85.5±8.1 |

Demographics and subjective sleep data are presented as means ± standard deviation. MMSE, Mini Mental State Examination; PSQI, Pittsburg Sleep Quality Inventory; WASO, Wake after sleep onset; REM, Rapid eye movement; SE, Sleep efficiency.

All subjects were cognitively healthy (MMSE ≥ 27) and well-educated (14 to 20 years of education). The PSQI total score ranged from 0 to 13 (<5 'Good sleep quality', ≥5 'Poor sleep quality'), with 78% of the sample (25/32) reporting good sleep quality. Most of the sample showed low risk for sleep apnea as assessed using the Berlin Sleep Questionnaire (81.25%).

**b) Linear mixed models correlating subjective and objective sleep parameters.** Detailed coefficients from the linear mixed models examining the association between PSQI components and objective sleep measures are shown in Table 2. The table shows the effect of Zeo parameters on PSQI components using a mixed effects linear model adjusted by age, gender, Education and likelihood of sleep apnea (as determined with the Berlin Sleep Questionnaire).

**Table 2. Linear mixed effects models of PSQI components and Zeo parameters.**

| PSQI subcomponent | Sleep Disturbance | | Daytime Disfunction | | Sleep Efficiency | |
|---|---|---|---|---|---|---|
| Zeo Parameter models | β (SE) | p-value | β (SE) | p-value | β (SE) | p-value |
| **WASO (%)** | **1.06 (0.04)** | **0.005** | 0.12 (0.03) | 0.741 | -0.50 (0.04) | 0.202 |
| **WASO SD (%)** | **-0.72 (0.08)** | **0.042** | -0.93 (0.04) | 0.793 | 0.52 (0.05) | 0.517 |
| **WASO (Min)** | **0.69 (0.14)** | **0.010** | 0.09 (0.00) | 0.730 | -0.22 (0.01) | 0.423 |
| **WASO SD (Min)** | -0.49 (0.21) | 0.069 | -0.27 (0.01) | 0.916 | 0.29 (0.01) | 0.281 |
| **SE (%)** | **-0.82 (0.23)** | **0.016** | -0.26 (0.02) | 0.937 | **0.67 (0.15)** | **0.047** |
| **SE SD (%)** | -0.40 (0.17) | 0.186 | 0.04 (0.04) | 0.902 | **0.69 (0.11)** | **0.036** |
| **TST (Min)** | **-0.40 (0.09)** | **0.030** | -0.06(0.00) | 0.713 | 0.16 (0.00) | 0.386 |
| **TST SD (Min)** | -0.13 (0.08) | 0.488 | -0.04 (0.00) | 0.807 | 0.20 (0.01) | 0.201 |
| **Deep Sleep (Min)** | -0.25 (0.00) | 0.265 | **-0.43 (0.11)** | **0.027** | -0.04 (0.01) | 0.858 |
| **Deep Sleep SD (Min)** | -0.22 (0.01) | 0.367 | 0.07 (0.08) | 0.741 | 0.27 (0.02) | 0.914 |

Data are presented as regression coefficients and standard errors (SE). This table shows the effect of Zeo parameters on PSQI components using a mixed effects linear model adjusted by age, gender, MMSE, Education and Berlin Apnea Index with a random effect for events with subjects.

None of the objective sleep measures had a significant association with PSQI total score (RS (%) p = 0.827, DS (%) p = 0.711, LS (%) p = 0.458, WASO (%) p = 0.515, RS (min) p = 0.470, DS (min) p = 0.203, LS (min) p = 0.997, WASO (min) p = 0.545, SE p = 0.370, TST = 0.833) but showed significant relationships with PSQI components. Models were fit separately to groups of Zeo parameters because of the small sample size.

Among the PSQI components, objective SE and PSQI SE (β = 0.67, $p<0.047$) were positively associated. Objective WASO (% and min) and PSQI sleep disturbance were positively associated, where the greater the WASO, the higher the PSQI sleep disturbance score (WASO % β = 1.06, $p$ = 0.005; WASO Min β = 0.69, $p$ = 0.01). We also found that objective SE (% β = -0.82, $p$ = 0.016) was negatively associated with PSQI sleep disturbance, where the less efficient the sleep, the greater the sleep disturbance. Similarly, objective TST (Min β = -0.40, $p$ = 0.03) was negatively associated with PSQI sleep disturbance, where the less time the participant slept, the greater the sleep disturbance. No other significant associations were found with PSQI sleep disturbance.

Lastly, we found a significant negative association between objective Deep Sleep (min) and PSQI Daytime dysfunction (β = -0.43, $p$ = 0.027), where the less Deep Sleep the more Daytime dysfunction. The standard deviation (SD) of objective sleep parameters was significant only in two models: 1.- when evaluating the WASO effect on PSQI Sleep Disturbances (β = -0.72, $p$ = 0.042) and 2.- when looking at the objective SE effect on PSQI SE (β = 0.69, $p$ = 0.036). In other words, the WASO variability across nights was negative, highlighting the worsening effect on PSQI Sleep Disturbance (showing higher precision in this negative effect) and the SE variability across nights made the PSQI SE more highly variable and less efficient.

## 4. Discussion

We hypothesized that SOL and SE as measured by the ZEO device, would be associated with both PSQI total score and components, however, our results showed that objective SE, WASO and TST, but not SOL, are significantly associated with only the sleep disturbance component of the PSQI.

While subjective and objective sleep assessments are often incongruent, our goal was to determine if the subjective measures were driven by a specific aspect of objective sleep. For example, subjective feelings of how someone sleeps (captured by the PSQI) may be driven by how long it took the person to fall asleep (SOL) as opposed to how much someone slept overall (TST). Researchers agree that self-estimation of sleep by specific questionnaires are useful methods for assessing subjective sleep and sleep disturbances in large samples. The PSQI yields, in less than 10 questions, information about several measurements of sleep, including SOL, SE, TST, and sleep quality or satisfaction, which should reflect a subjective global appraisal of several nights of sleep [29].

The first interesting finding in our study was that objective measures might be related only to certain components of the scale, and while we tend to report just the total score, it could be that some questions better represent our objective sleep than others, so if we only look at the global score, we can miss such a relationship. In this case, only PSQI Sleep disturbance, SE and Daytime dysfunction were representative of the objective findings. Specifically, percentage of WASO and total minutes of WASO were, according to our objective measure, positively correlated with PSQI sleep disturbance. This association does not surprise us, given that the more time spent awake during the night the more disturbed your sleep will feel. Following the same logic, it is expected that lower objective SE and TST will consequently make our sleep subjectively feel more disturbed.

Lastly, we found that the more PSQI Day Dysfunction, the less average minutes in deep sleep. One possible explanation of such finding is that the least number of minutes you spend

in deep sleep, the sleepier you are, which is reported as Daily Dysfunction on the PSQI. However, based on all of the other measures attained, PSQI does not otherwise reflect measures of sleep stages but instead overall sleep amounts.

In its original paper, Buysse et al. discussed the lack of agreement between the PSQI total score and polysomnographic variables, arguing that the PSQl asks for a global estimate spanning 1 month, and is not sensitive to daily variability [11]. However, we used an objective sleep measure that spans 10 days, as opposed to the 1–2 usually measured by the laboratory PSGs, which allowed us to address such concerns. Furthermore, we assessed both the mean and standard deviation of each sleep variable across the acquired sleep study nights in order to analyze night-to-night sleep variations and see if this contributed to the perceived sense of poor sleep quality, but we did not find SD (night-to-night variability) to be statistically significant. We are aware of only one study that tried to quantify the subjective sleep quality in healthy elderly men and women by focusing on the PSQI components and not just the whole score [14]. They compared older adults to young adults, and showed worse PSQI component scores for subjective sleep quality, sleep duration, SE, and sleep disturbances in older adults [14]. Interestingly, most subjects from their cohort of healthy older adults rated their subjective sleep as "good" and argued that, even though this does not correlate with the objective data, the gradual changes in sleep quality associated with aging may lead the older individual to adapt their perception of sleep quality to the actual changes in their sleep structure, and thus not recognize that their typical sleep is actually disrupted [16]. In agreement with this theory, Madrid-Valero et al. showed worsening of sleep with advancing age, affecting SOL and SE and increasing the prevalence of sleep disorders, but once again, did not find significant differences in the perception of sleep quality, as measured by the PSQI total scores [30].

Parsey et al. also correlated the PSQI to objective actigraphy data, and in contrast to our findings, did not find a significant correlation between any subjective and objective measure [21]. Similarly, Landry et al. concluded that PSQI does not provide predictive validity for an older adult's objective sleep quality; and thus, should be used with caution [19]. These differences might be explained by the different objective measures used between the studies: while we used a wearable frontal EEG monitor, they used actigraphy, which might over-estimate sleep by counting the moments in which the individual is quietly resting in bed, as sleep [21]. In any case, we believe the PSQI can provide meaningful information, but it might not be as simple as the total score.

Using different scales from the PSQI, Woosley et al. reported that among women, the number of awakenings, WASO and TST frequently predicts self-reported sleep quality [31]. Their findings are relevant to our study, not only because they are consistent with ours, but also because it proves that the correlation might be better measured by specific components of the different scales, and not by the total score [32].

According to Driscoll et al, good sleep may be a marker of good functioning across a variety of domains in old age, highlighting the need to study interventions which protect sleep quality [33]. The above results suggest that these interventions should not be limited to shortening the SOL -which they often do- but more importantly, focus on maintaining sleep (reducing WASO). This is particularly relevant as many of the SOL-targeting medications can worsen sleep apnea, cause memory problems and daytime sleepiness.

Various sleep-related parameters, including long SOL, poor SE or poor quality, were reported to be associated with cognitive impairments in late life [34, 35]. However, findings on these associations have been inconsistent, probably due to the range in measuring tools used in these studies. While subjective tools are easier to apply and more cost-effective, they might not always be true to their nature. Interestingly, a study by Biddle et al., that examined whether poor objective (actigraphy) and subjective (Insomnia Severity Index) sleep quality are

differentially associated with cognitive function, found that only objectively measured poor SE may be associated with worse cognitive function (independent of depression severity), while subjective sleep quality was not related to cognitive function [36]. These highlight the importance of using the right tool to measure sleep quality.

Some strengths and limitations are important to note. First, we were able to perform the sleep study in a real-life setting (home environment) for 10 days, enabling analysis of sleep variability. Second, we were able to utilize frontal EEG and not rely solely on wrist actigraphy. Third, the mobile frontal EEG device allowed assessment of the relationship between PSQI with summarized sleep stages. Four, few sleep studies have focused on clinically confirmed cognitively healthy older adults as in this study; we believe this study on healthy, community-dwelling adults should be a valuable reference and prove useful in comparisons with specific patient populations, helping us to understand the degree of sleep disturbances in these specific populations. However, our overall study population was small, highly educated and racially homogenous, which may limit the applicability of our findings. Nevertheless, MMSE was used as a covariate and it did not change the results. Second, our estimates may be biased due to the lack of data for variables such as diagnosis of cardio-vascular disease and nocturia, however, this is a cohort of overall healthy adults, meaning this should not be an issue. Third, four of our participants were taking sleep aids at the time of study and seven were on medications whose side effect profile could affect sleep (e.g., SSRIs). Fourth, it could be argued that WASO is influenced by apnea, and apnea might be the real factor influencing sleep quality, however we controlled for the likelihood of sleep apnea, and this did not change our results. Fifth, as SOL was measured from the time the headband was put on and not from the moment the participant actually intended to sleep, we cannot be sure of its actual value and some might consider that Zeo measured SOL is not an objective measure, however, the results regarding SOL were not significant. Participants in this study were instructed specifically on this, and told not to put the headband on until they were ready to try to fall asleep. Sixth, the ZEO device does not provide raw EEG signals. Lastly, the present study was conducted as a cross-sectional analysis of older adults; future longitudinal studies are the next step to better understand these determinants of sleep quality and improve the prediction of perceived sleep quality.

Results from this study highlight that low-cost tools can be beneficial in clinical settings, particularly when looking at component scores rather than just a total summary score. Using PSQI can help identify cognitively healthy older adults with issues with sleep onset vs sleep maintenance, modifying the prescribed therapy and hopefully diminishing potential sleep-related cognitive decline.

## Acknowledgments

The authors thank the participants for their invaluable contribution to brain aging research. We also thank all of the research coordinators, sleep technicians, neurologists and nursing staff who helped with the data collection towards this study.

## Author Contributions

**Conceptualization:** Jennifer Zitser, Christine M. Walsh.

**Data curation:** Michael M. Le.

**Formal analysis:** Jennifer Zitser, Isabel Elaine Allen, Neus Falgàs.

**Funding acquisition:** Thomas C. Neylan, Joel H. Kramer, Christine M. Walsh.

**Methodology:** Christine M. Walsh.

**Resources:** Thomas C. Neylan.

**Writing – original draft:** Jennifer Zitser.

**Writing – review & editing:** Isabel Elaine Allen, Neus Falgàs, Thomas C. Neylan, Christine M. Walsh.

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
