## [Decision Letter · Decision Letter 0]

3 Feb 2022

PONE-D-21-37989Pittsburgh Sleep Quality Index (PSQI) Responses are Modulated by Total Sleep Time and Wake After Sleep Onset in Healthy Older AdultsPLOS ONE

Dear Dr. Zitser Koren,

Thank you for submitting your manuscript to PLOS ONE. After careful consideration, we feel that it has merit but does not fully meet PLOS ONE’s publication criteria as it currently stands. Therefore, we invite you to submit a revised version of the manuscript that addresses the points raised during the review process.

We look forward to receiving your revised manuscript.

Kind regards,

Federica Provini

Academic Editor

PLOS ONE

Journal Requirements:

(This   study was supported by UCSF Hillblom Aging Network grant, Clinical features and neuropathological basis of sleep-wake behavior in Alzheimer’s Disease and PSP (NIA R01 AG060477), Linking sleep dysfunction to tau-related degeneration across AD progression (NIA R01 AG064314), Biological predictors of brain aging trajectories (NIA R01 5R01AG032289-10) and the Tau Consortium/Rainwater Charity Foundation.)

Reviewers' comments:

Reviewer's Responses to Questions

**Comments to the Author**

1. Is the manuscript technically sound, and do the data support the conclusions?

Reviewer #1: Partly

2. Has the statistical analysis been performed appropriately and rigorously? 

Reviewer #1: No

3. Have the authors made all data underlying the findings in their manuscript fully available?

Reviewer #1: No

4. Is the manuscript presented in an intelligible fashion and written in standard English?

Reviewer #1: Yes

5. Review Comments to the Author

Reviewer #1: The objective of this paper is to identify the objective sleep influencers to subjective sleep measure, PSQI. They found certain objective sleep parameters were not associated with PSQI total score, but with PSQI component score of sleep disturbance. Considering that the prediction of objective sleep parameters by PSQI total or component score would be another way of looking into the usefulness of PSQI, there is a limitation in interpreting the results of this study.

There are several concerns in this paper as follows:

1) Statistical analysis

- In the sentence of “A random effect for events within subjects was used as subjects contributed a varying number of sleep events”, the meaning of sleep events needs to be described in detail.

- Using linear mixed model analysis in this study, objective sleep measures and their standard deviations were included together in the model. Is it acceptable that this is due to small sample size, or is there any other possibility of applying a different analysis for assessing the variability of these measures? Please provide a statistician’s view on whether this analysis is appropriate.

2) Results

- In table 2, there might be an error in the beta (-0.69) of WASO (min). The description on this in the text is different from that of the table.

- In lines 279-281, the interpretation of the result on SE variability is not correct. It needs to be described differently.

3) Discussion

- There are some contents which are not relevant to the results of this study. (i.e, lines 348-356). Please reduce the length of the discussion by excluding the unnecessary parts in order to focus on the essential points of this study.

- Since the result in lines 320-321 was not included in the table 2, a statistical result regarding the description should be included.

- Among some strengths of this study described, the assessment of sleep stages by the mobile frontal EEG device might not be relevant.

- Zeo measured SOL might be inaccurate due to its technique. The discussion about the relationship of SOL with PSQI should be mentioned, although there was no significant result on this.

6. PLOS authors have the option to publish the peer review history of their article (what does this mean?). If published, this will include your full peer review and any attached files.

Reviewer #1: No

---

## [Author Response · Author response to Decision Letter 0]

17 May 2022

We would like to thank the reviewers for carefully reading our manuscript and for the many insightful comments and suggestions. We believe that the reviewers’ comments have strengthen our paper and for that we are grateful. Please find our responses to the reviewers’ comments below. 

Reviewer #1: The objective of this paper is to identify the objective sleep influencers to subjective sleep measure, PSQI. They found certain objective sleep parameters were not associated with PSQI total score, but with PSQI component score of sleep disturbance. Considering that the prediction of objective sleep parameters by PSQI total or component score would be another way of looking into the usefulness of PSQI, there is a limitation in interpreting the results of this study.

There are several concerns in this paper as follows:

1) Statistical analysis

- In the sentence of “A random effect for events within subjects was used as subjects contributed a varying number of sleep events”, the meaning of sleep events needs to be described in detail.

The “events” are the number of nights contributed by each participant - variable number of nights (which in a linear model we call events) so we need a random effect for the total number of nights within each participant. However, the wording was indeed confusing, which is why we have replaced it with “nights” instead of “sleep events”.

- Using linear mixed model analysis in this study, objective sleep measures and their standard deviations were included together in the model. Is it acceptable that this is due to small sample size, or is there any other possibility of applying a different analysis for assessing the variability of these measures? Please provide a statistician’s view on whether this analysis is appropriate.

As requested by the reviewers, this reply is being provided by Isabel Elaine Allen, PhD Professor of Biostatistics & Epidemiology at UCSF: 

Sample size is not an assumption guiding the use of linear mixed models as the analysis incorporates the variability and the number of events included from each patient. For a relatively small sample size the variability will be greater than with a larger sample size but this is reflected in the models. It is a good way to assess variables when a different number of events (in this case, number of nights) are contributed by the participants because this is controlled for in the model with participants having a small number of events receiving less weight.

2) Results

- In table 2, there might be an error in the beta (-0.69) of WASO (min). The description on this in the text is different from that of the table.

 We thank the reviewers for spotting this mistake. There is in fact an error in Table 2 and it should be 0.69, so the description is correct and the result in the table has been changed.

- In lines 279-281, the interpretation of the result on SE variability is not correct. It needs to be described differently.

The interpretation had been paraphrased as follows:

“In other words, the WASO variability across nights was negative, highlighting the worsening effect on PSQI Sleep Disturbance (showing higher precision in this negative effect) and the SE variability across nights made the PSQI SE more highly variable and less efficient.”

3) Discussion

- There are some contents which are not relevant to the results of this study. (i.e, lines 348-356). Please reduce the length of the discussion by excluding the unnecessary parts in order to focus on the essential points of this study.

We have shortened the paragraph.

- Since the result in lines 320-321 was not included in the table 2, a statistical result regarding the description should be included.

 P values showing that these results were not significant were added under the “Results” section, “Linear mixed models correlating subjective and objective sleep parameters” subheading (line 255). It now reads like this”:

“None of the objective sleep measures had a significant association with PSQI total score but showed significant relationships with PSQI components.”

- Among some strengths of this study described, the assessment of sleep stages by the mobile frontal EEG device might not be relevant.

Even though the reviewer is right in that the individual sleep stages might not be relevant for this study, having the EEG to better discern sleep Vs wake is better than relying purely in actigraphy, which is why we believe that this is something worth mentioning.

- Zeo measured SOL might be inaccurate due to its technique. The discussion about the relationship of SOL with PSQI should be mentioned, although there was no significant result on this.

We are aware of Zeo’s limitation regarding SOL, and as such we mentioned it at the end of our discussion (line 388). We have added an extra sentence commenting on the lack of significant results.

Thank you again for your comments, we hope we have addressed all of your concerns.

---

## [Editor Report · Decision Letter 1]

6 Jun 2022

Pittsburgh Sleep Quality Index (PSQI) Responses are Modulated by Total Sleep Time and Wake After Sleep Onset in Healthy Older Adults

PONE-D-21-37989R1

Dear Dr. Koren,

We’re pleased to inform you that your manuscript has been judged scientifically suitable for publication and will be formally accepted for publication once it meets all outstanding technical requirements.

Kind regards,

Federica Provini

Academic Editor

PLOS ONE

---

## [Editor Report · Acceptance letter]

15 Jun 2022

PONE-D-21-37989R1 

Pittsburgh Sleep Quality Index (PSQI) Responses are Modulated by Total Sleep Time and Wake After Sleep Onset in Healthy Older Adults 

Dear Dr. Zitser Koren:

I'm pleased to inform you that your manuscript has been deemed suitable for publication in PLOS ONE. Congratulations! Your manuscript is now with our production department. 

Kind regards, 

on behalf of

Dr. Federica Provini 

Academic Editor

PLOS ONE